# Performance of immunological assays for universal and differential diagnosis of HTLV-1/2 infection in candidates for blood donations from the Brazilian Amazon

Felipe Araujo Santos[1,2,3], Cláudio Lucas Santos Catão[1,2], Júlia Pereira Martins[4,5], Uzamôr Henrique Soares Pessoa[2], Isabelle Vasconcelos Sousa[2,6], Jean Silva Melo[3,7], Gláucia Lima Souza[1,2], Nilberto Dias Araújo[1,2,3], Fábio Magalhães-Gama[2,4,5], Cláudia Maria de Moura Abrahim[2], Emmily Myrella Vasconcelos Mourão[7], Vanessa Peruhype-Magalhães[5], Jordana Grazziela Alves Coelho-dos-Reis[5,8], Andréa Teixeira-Carvalho[4,5], Antonio Carlos Rosário Vallinoto[9,10], Gemilson Soares Pontes[1,3,7], Márcio Sobreira Silva Araújo[4,5], Olindo Assis Martins-Filho[1,4,5], Allyson Guimarães Costa[1,2,3,6]*

1 Programa de Pós-Graduação em Ciências Aplicadas à Hematologia, Universidade do Estado do Amazonas (UEA), Manaus, Brazil, 2 Diretoria de Ensino e Pesquisa, Fundação Hospitalar de Hematologia e Hemoterapia do Amazonas (HEMOAM), Manaus, Brazil, 3 Programa de Pós-Graduação em Imunologia Básica e Aplicada, Instituto de Ciências Biológicas, UFAM, Manaus, Brazil, 4 Programa de Pós-Graduação em Ciências da Saúde, Instituto René Rachou—Fundação Oswaldo Cruz (FIOCRUZ) Minas, Belo Horizonte, Brazil, 5 Grupo Integrado de Pesquisas em Biomarcadores, Instituto René Rachou—FIOCRUZ Minas, Belo Horizonte, Brazil, 6 Programa de Pós-Graduação em Enfermagem, Escola de Enfermagem de Manaus, Universidade Federal do Amazonas (UFAM), Manaus, Brazil, 7 Laboratório de Virologia, Instituto Nacional de Pesquisa da Amazônia (INPA), Manaus, Brasil, 8 Departamento de Microbiologia, Instituto de Ciências Biológicas, Universidade Federal de Minas Gerais (UFMG), Belo Horizonte, Brazil, 9 Programa de Pós-graduação em Biologia de Agentes infecciosos e Parasitários, Universidade Federal do Pará (UFPA), Belém, Brasil, 10 Laboratório de Virologia, Instituto de Ciências Biológicas, Universidade Federal do Pará, Belém, Brazil

* allyson.gui.costa@gmail.com

**Data Availability Statement:** The original contributions presented in the study are included in

## Abstract

The present study compares the ability of distinct immunological assays (chemiluminescence immunoassay-CLIA, western blot-WB and flow cytometry-FC-Simplex and Duplex) to detect anti-HTLV (human T-lymphotropic virus) antibodies in candidates for blood donations at the Amazonas State Blood Center (Brazil) between January 2018 and December 2022. Overall, 257,942 samples from candidates for blood donations were screened using CLIA, which led to 0.15% seropositivity for HTLV (409 samples). A total of 151 candidates for blood donations were enrolled for retesting with CLIA followed by additional testing using WB and FC-Simplex and Duplex analysis. Our results demonstrated that 62% (93/151), 20% (30/151) and 17% (26/151) of the samples presented positive results with retesting using CLIA, WB and FC-Simplex analysis, respectively. Additional analysis of the CLIA, WB and FC-Simplex results revealed an overall agreement of 56% for CLIA and WB (22 co-negative; 30 co-positive samples), 48% for CLIA and FC-Simplex (21 co-negative; 24 co-positive samples) and 80% for WB and FC-Simplex (51 co-negative; 23 co-positive samples). Considering the WB as the reference standard for the diagnosis of infection with HTLV-1/2, we observed that the CLIA results of ≤3.0 RLU and >10.0 RLU in the retest can be used

the article/supplementary material, further inquiries can be directed to the corresponding author/s. Due to ethical restrictions regarding patient privacy, data are available upon request. Data are available upon request from the Ethics Committee of HEMOAM (CEP-HEMOAM), cep@hemoam.am. gov.br, for researchers who meet the criteria for access to confidential data. Additional requests for the data may be sent to the corresponding author or coauthors Allyson G. Costa (allyson.gui. costa@gmail.com) and Olindo A. Martins-Filho (oamfilho@gmail.com).

**Funding:** Financial support was provided in the form of grants from Fundação de Amparo à Pesquisa do Estado do Amazonas (FAPEAM) (Pró-Estado Program #002/2008, #007/2018 and #005/ 2019, PRODOC Program #003/2022; STARTUP PARA O SUS Program #012/2022 and POSGRAD Program #002/2023 and #002/2024), Fundação de Amparo à Pesquisa do Estado de Minas Gerais (FAPEMIG) (APQ-00821-20 – PPSUS Program #03/2020), Conselho Nacional de Desenvolvimento Científico e Tecnológico (CNPq) and Coordenação de Aperfeiçoamento de Pessoal de Nível Superior (CAPES) (PROCAD-Amazônia 2018 Program - #88881.200581/2018-01). FAS, CLSC, JPM, IVS and GLS have fellowships from FAPEAM, FAPEMIG and CAPES (MSc students). JSM was research fellow from CAPES (PhD student). UHSP have fellowships from FAPEAM (scientific initiation student). JGAC-R, ACRV, MSSA, OAM-F and AGC are research fellows from CNPq. OAM-F participated in the fellowship program supported by the Universidade do Estado do Amazonas (PROVISIT N° 005/2023-PROPESP/UEA). The funders had no role in study design, the decision to publish, or preparation of the manuscript.

**Competing interests:** The authors have declared that no competing interests exist.

define a negative or positive result, respectively, and could be used as new specific cut-off values. The overall agreement between WB and FC-Duplex for accomplishing the differential diagnosis was evaluated and demonstrated 100% correspondence for the diagnosis of HTLV-1 (15/15) and HTLV-2 (7/7). Our findings demonstrate that gaps in the diagnosis of infection with HTLV-1/2 could be overcome by the simultaneous use of distinct immunological assays during retesting of candidates for blood donations.

## 1. Introduction

Discovered in 1980, human T-lymphotropic virus (HTLV) is an oncogenic retrovirus that has 4 types (HTLV-1, HTLV-2, HTLV-3 and HTLV-4). HTLV-1 and HTLV-2 are viruses that are considered to be of medical importance and are related to adult T-cell leukemia/lymphoma (ATLL), neurodegenerative diseases such as HTLV-1 associated myelopathy/tropical spastic paraparesis (HAM/TSP) and hairy cell infections [1–3]. It is estimated that 3 to 5% of HTLV-1 infections may evolve into these pathologies, while HTLV-2 infection still needs to be better characterized regarding its relationship with these clinical diseases [4–10].

Transmission of the virus in humans can occur through sexual contact, blood transfusions, transplants with contaminated organs, breastfeeding of newborns or accidents with or sharing of contaminated sharp instruments [7, 11–13]. The identification of people living with HTLV (PLHTLV) is essential for multidisciplinary follow-up of patients and asymptomatic carriers, thus reducing the medium- and long-term impacts through the emergence of diseases associated with HTLV [14, 15]. In addition, HTLV-1 infection is associated with inflammatory diseases such as uveitis, conjunctivitis, Sicca syndrome, interstitial keratitis, infective dermatitis, arthritis, myositis, Sjögren's syndrome, Hashimoto's thyroiditis, Graves' disease and polyneuropathies, and is also associated with the aggravation of other infectious diseases such as tuberculosis and strongyloidiasis [16].

HTLV-1/2 infections are estimated to affect approximately 10 to 20 million people worldwide and depending on the region, prevalence can range between 5% and 27% [1, 17]. Areas of Japan, the Caribbean Islands, Africa, South America, some regions of Romania and the Middle East are endemic for HTLV-1, while HTLV-2 is found in pygmies in Central America, African countries, and indigenous populations of the Americas [1, 2, 18]. In Brazil, the first cases of HTLV were detected in 1986, in a community of Japanese descendants in Mato Grosso do Sul [19].

Subsequently, the detection of the virus in different Brazilian regions and states was described, with a greater number of cases in the states of Bahia, Maranhão and Pará, although the infection is considered endemic and has a heterogeneous distribution [20, 21]. In the northern region of Brazil, reports have shown a high prevalence of HTLV-1/2 infection in blood donors from the states of Amapá (0.71%) and Pará (0.91%), although in Amazonas the number of cases is considered low (0.13%) [22–24].

Due to their higher sensitivity, the screening of patients or blood donors for HTLV-1/2 is initially carried out using the enzyme immunosorbent assay (ELISA), chemiluminescent immunoassay (CLIA) and the particle agglutination assay (PA). Subsequently, confirmatory assays are performed using western blot (WB), line immuno assay (INNO-LIA) and quantitative real-time polymerase chain reaction (qPCR). These assays have a high specificity for detecting specific antibodies for different HTLV antigens, as well as molecular detection of the genetic material of the provirus [25, 26]. In addition to these tests, a new method for screening

and differential diagnosis of HTLV-1/2 infection based on the flow cytometry technique presented promising results with the Flow Cytometry (FC)-Simplex IgG1 (HTLV) and FC-Duplex IgG1 (HTLV-1/2) assay, respectively [27–29].

WB is considered to be the reference standard for the detection and diagnosis of HTLV-1/2 infection, although it has limitations regarding seroconversion in newly infected individuals, in addition to high costs due to the limited availability of tests [30–32]. Moreover, the difficulty of automating these assays is noted, with the use of CLIA being proposed for blood centers since it has excellent sensitivity and the ability to track infection, despite false-positive results [31, 33–35]. Thus, a study was carried out in order to compare the performance of distinct immunological assays (CLIA, WB, FC-Simplex and Duplex) to detect anti-HTLV antibodies in candidates for blood donations at the Amazonas State Blood Center (Brazil).

## 2. Materials and methods

### 2.1 Study design

A longitudinal study was carried out with blood donor candidates from the state of Amazonas, Brazil, who were evaluated at the Fundação Hospitalar de Hematologia e Hemoterapia do Amazonas (HEMOAM) and who had a positive result for HTLV-1/2 infection via the CLIA in the period from January 1, 2018, to December 31, 2022. The state of Amazonas has a low prevalence of HTLV-1/2 infection, which was previously estimated to be 0.13% [23].

### 2.2 Ethical statement

Ethical approval for the study was obtained from the Research Ethics Committee at the HEMOAM Foundation (approval number #5.348.608/2021 and #57153922.5.0000.0009). All procedures are in accordance with Resolution 466/12 of the Brazilian Ministry of Health and the Declaration of Helsinki. All the participants read and signed the informed consent form (ICF) before enrollment.

### 2.3 Characteristics of the study population

Individuals aged 18 years or older who were seropositive for anti-HTLV-1/2 in the CLIA screening test were interviewed in the period from February 1 to October 31, 2023. Those who accepted and signed the ICF were subjected to a CLIA retest, in addition to the WB, FC-Simplex and FC-Duplex assays. The results of immunological assays were not used for medical decisions regarding diagnosis and treatment. Furthermore, participants who tested positive for HTLV-1 or HTLV-2 infection via the WB assay were considered to have a defined diagnosis.

### 2.4 Acquisition of sociodemographic, clinical and laboratory data

Demographics and laboratory information were collected in the period from January 15 to September 30, 2023 from the records of the Serology Laboratory using the HEMOSYS system. Sociodemographic and clinical information were collected during the application of the questionnaire and collection of biological samples.

### 2.5 Collection and processing of biological samples

Venous blood for immunological assays was collected using a vacuum system in tubes with EDTA (ethylenediaminetetraacetic acid) and separating gel (BD Vacutainer EDTA® K2 and BD gel SST® Advance II). Subsequently, the biological samples were centrifuged, and aliquots of ±700 uL were prepared and frozen at -80˚C until the CLIA, WB, FC-Simplex and FC-Duplex assays were carried out (**S1 File**).

## 2.6 Chemiluminescence assay (CLIA)

The chemiluminescence assay (CLIA) was carried out in the serology sector of the HEMOAM Foundation as a screening and retesting step to detect donors that were positive for HTLV-1/2 infection. The procedure consists of a two-step immunoassay for the qualitative detection of antibodies to HTLV-1 and HTLV-2 in serum, using the Alinity's rHTLV-I/II kit (Abbott®). The immunoassay is performed with a closed system and chemiluminescent microparticle technology (Alinity's i-SCM 02 Ai01767, Abbott®). The tests were carried out following the guidelines of the kit and equipment's manufacturers, with results expressed in relative light units (RLUs). According to the criteria of the Alinity's rHTLV-I/II kit, a sample with RLUs >1.0 was considered reactive for HTLV-1/2 antibodies.

## 2.7 Western blot assay

The western blot (WB) assay was performed with the HTLV blot 2.4 kit (MP Diagnostics®), following the guidelines of the manufacturer of the kit. The WB is a qualitative enzyme immunoassay used for the detection of antibodies to HTLV-1/2 in human serum and is used as a confirmatory step that is capable of detecting and distinguishing between types 1 and 2 of the virus, as recommended by the Food and Drug Administration (FDA) and Brazilian Ministry of Health [25, 36]. Considering the band profiles, the WB results were classified into three categories: negative (no reactivity or reactivity to proteins other than p19 or p24), indeterminate (reactivity to specific HTLV bands that do not meet the criteria for HTLV-1/2 seropositivity) or positive results (reactivity to Gag (p19 with or without p24) and two Env proteins (GD21 and rgp46-I) for HTLV-1 or Gag (p19 with or without p24) and two Env proteins (GD21 and rgp46-II) for HTLV-2.

## 2.8 Flow cytometry assay

The FC-Simplex IgG1 (HTLV) assay was developed as a qualitative support methodology for the detection of infections with HTLV, and uses commercial cells infected with the virus [27, 29]. This technology uses a system for detecting IgG1 antibodies on a competitive immunofluorescence platform using flow cytometry, with MT-2 cells as antigenic support. The assay aims to select positive samples, but without distinguishing between viral types, and functions as a screening step for FC-Duplex IgG1 HTLV. Subsequently, the FC-Duplex IgG1 (HTLV-1/2) assay was performed as a supporting assay for the differential diagnosis of HTLV-1/2 infection, using commercial cells infected with the viruses [28]. The assay uses a system for detecting IgG1 antibodies in a competitive immunofluorescence platform using flow cytometry, using MT-2 and MoT cell lines stained with the fluorochrome Alexa Fluor 647 (AF647) as antigenic support. In both tests, the results are expressed as a percentage of positive fluorescent cells (PPFC), which were obtained with the detection system after acquiring the samples in the FACSCalibur cytometer and analyzing the MFI with the FlowJo software (v10.1).

## 2.9 Statistical analysis

Prisma GraphPad Software version v.8.1 (GraphPad Prism, San Diego, CA, USA) was used to create graphs and perform the statistical analyses. The Shapiro-Wilk test was used to verify the distribution and normality of the variables, which showed a non-parametric distribution. The comparison of RLU and PFFC data was performed using the Mann-Whitney test, and the Spearman correlation coefficient was used to evaluate the relationship between RLU and PFFC data. The level of statistical significance in all the analyses was defined as $p < 0.05$. The accuracy test was calculated using an online application (https://www.openepi.com/

DiagnosticTest/DiagnosticTest.htm) and Stata software, version 16.0 (StataCorp LLC, College Station, TX, USA). Sensitivity was calculated as the proportion of participants with HTLV-1/2 who had a positive WB/CLIA/FC-Simplex test, and specificity was calculated as the number of participants with a negative WB/CLIA/FC-Simplex test result. The positive predictive value (PPV) was calculated as the proportion of participants among those with a positive WB/CLIA/FC-Simplex result, and the negative predictive value (NPV) as the proportion of participants with a negative WB/CLIA/FC-Simplex.

## 3. Results

During the study period (Jan, 2018 to Dec, 2022), 409 samples from the 257,942 candidates for blood donations (0.15%) presented a positive result for anti-HTLV-1/2 IgG during screening using the CLIA, which was carried out at HEMOAM. The median reactivity of the positive samples was 1.79 RLU. **Table 1** summarizes the demographical and laboratorial records of the candidates for blood donations with positive results for HTLV-1/2 during serological screening using the CLIA.

After the initial screening using the CLIA, donor candidates with positive results were invited to provide a new blood sample for retesting using the CLIA to confirm seropositivity for HTLV-1/2 antibodies (**Fig 1**). A total of 151 donor candidates agreed to participate in the retesting using the CLIA. Data analysis revealed that 93 out of the 151 samples (62%) exhibited positive results in the retest with the CLIA, with a median reactivity of 2.50 RLU (**Fig 1A**). Comparative analysis between positive CLIA results obtained during screening and the retest presented a high correlation "r" score (0.89, p<0.0001) (**Fig 1B**).

The 93 samples were additionally tested using two other immunological tests (WB and FC-Simplex). The results demonstrated that 30 samples (20%) presented positive results in the WB and 26 (17%) in the FC-Simplex (**S1 Table**). Additional analysis of the scattering distribution of the CLIA results of the retesting allowed the identification of three clusters, which were further classified as: low RLU (1.0–1.3), intermediate RLU (>1.3–3.0) and high RLU (>3.0–400), comprising 22%, 35% and 43% of tested samples, respectively. Data from the WB were also classified into three categories, considering the band profiles, as: negative (no reactivity or reactivity to proteins other than p19 or p24), indeterminate (reactivity to HTLV specific bands that do not meet the criteria for HTLV-1/2 seropositivity) and positive results (reactivity to p19, GD21 and rgp46-I for HTLV-1 or reactivity to p24, GD21 and rgp46-II for HTLV-2), comprising 57%, 11% and 32% of the tested samples, respectively. The results of FC-Simplex analysis were subsequently classified into two categories according to the magnitude of the PPFC results as: negative (PPFC≤20%) and positive (>20%) comprising 74% and 26% of the tested samples, respectively (**Fig 2A**).

**Table 1. Demographical and laboratorial records of candidates for blood donations with positive results for HTLV-1/2 during serological screening using CLIA.**

| Parameters | CLIA* screening of candidates for blood donations (n = 409) |
|---|---|
| **Age**, years (median [IQR]) | 33 [18–69] |
| **Gender** | |
| Male, n (%) | 209 (51%) |
| Female, n (%) | 200 (49%) |
| **CLIA* Reactivity** | |
| RLU Positive, (median [IQR]) | 1.79 [1.2–4.5] |

*CLIA: chemiluminescence assay; RLU: relative light units.

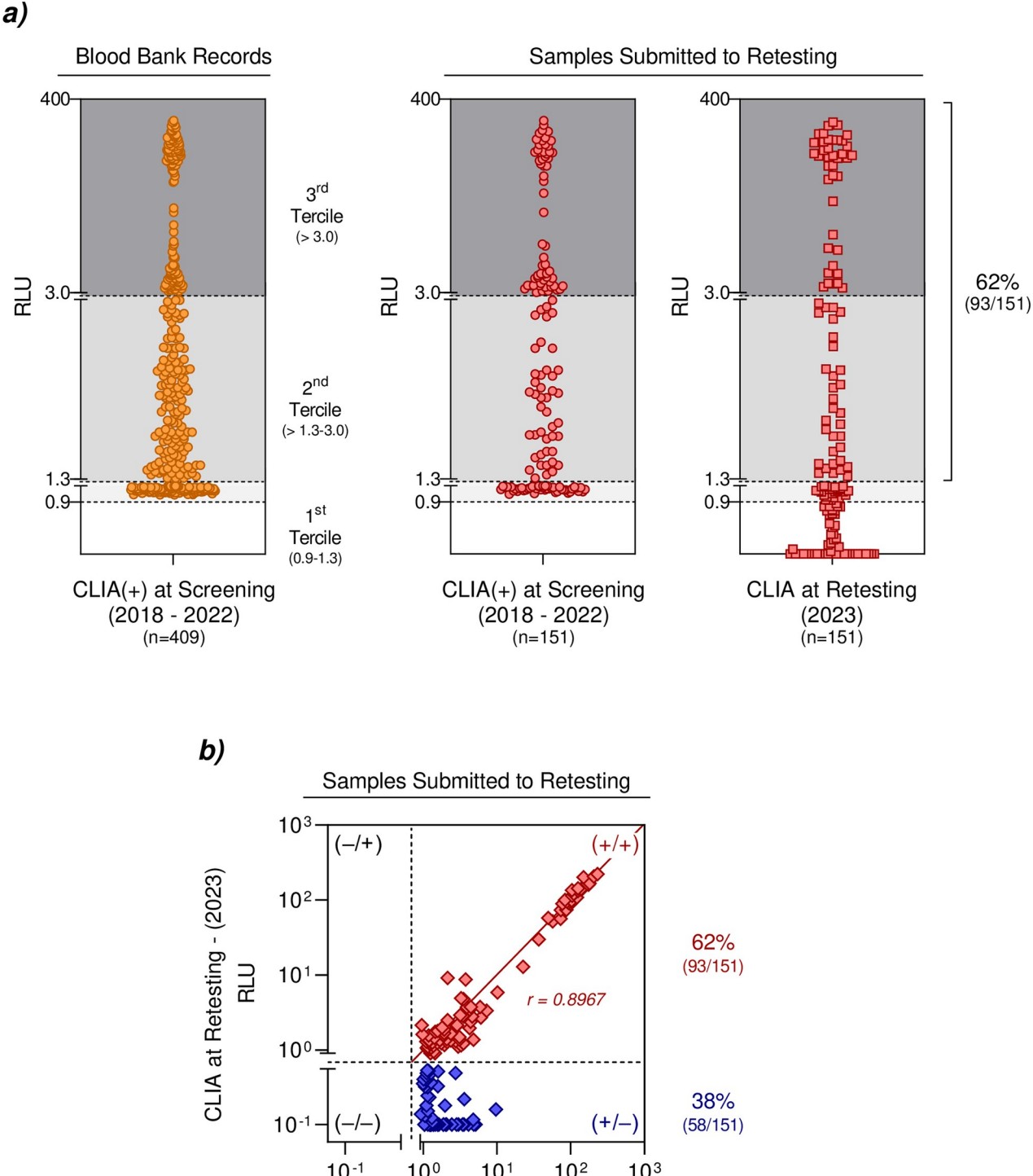

**Fig 1. Seroprevalence for anti-HTLV antibodies in candidates for blood donations at the Amazonas State Blood Center.** *a)* Anti-HTLV reactivity profile of 409 candidates for blood donations using CLIA during screening (2018–2022). CLIA results of 151 candidates for blood donations at retesting (2023). *b)* Correlation analysis between CLIA RLU values during screening and retesting. Statistical analyses were performed using the Mann-Whitney test for comparison of RLU and PFFC data, and Spearman correlation coefficient was used to evaluate the relationship between RLU and PFFC data. Significant differences (p<0.05). *CLIA: chemiluminescence assay; RLUs: relative light units.*

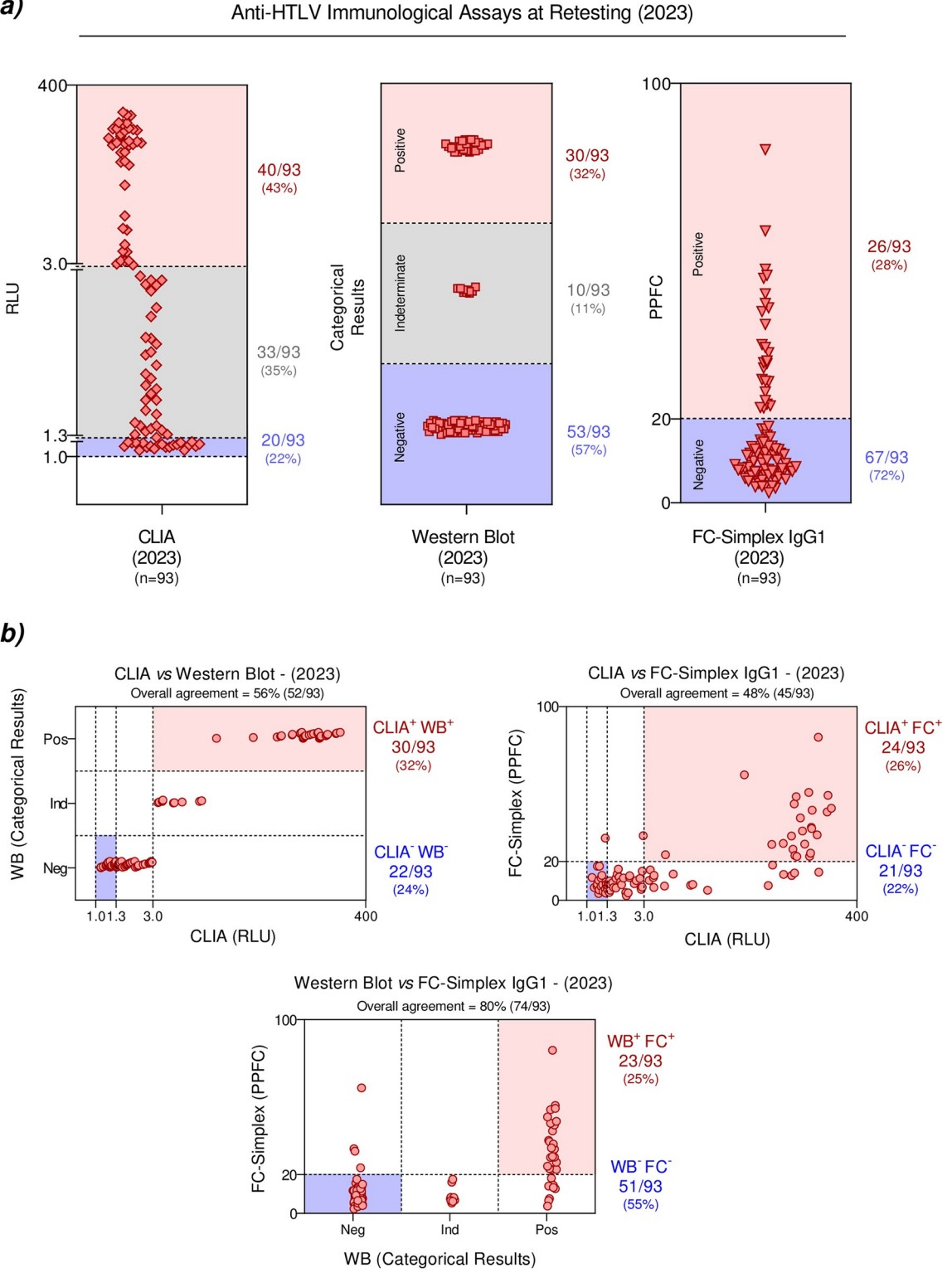

**Fig 2. Performance of different immunological assays to detect anti-HTLV antibodies.** *a)* CLIA results were classified as low, intermediate or high based on the RLU values; the western blot results were categorized as negative, indeterminate or positive, according to band profile, and FC-Simplex results were defined as negative or positive, according to the PPFC values. *b)* Comparison of CLIA vs WB, CLIA vs FC-Simplex and WB vs FC-Simplex results. *CLIA: chemiluminescence assay; RLU: relative light units; WB: western blot assay; PPFC: percentage of positive fluorescent cells.*

For further analysis of the results obtained during the retesting, the data from the CLIA, WB and FC-Simplex were analyzed considering the distinct categories of results obtained in each assay (low, intermediate and high—CLIA, positive, negative and indeterminate—WB, negative and positive—FC), as previously proposed in the **Fig 2A**. Comparative analysis of the CLIA and WB revealed an overall agreement of 56% (22 co-negative and 30 co-positive samples), while the CLIA and FC-Simplex yielded 48% of agreement (21 co-negative and 24 co-positive samples). Comparison between the WB and FC-Simplex demonstrated higher agreement (80%) with 51 co-negative and 23 co-positive samples (**Fig 2B**).

Using the WB as a reference method for confirmatory diagnosis of HTLV-1/2 infection, our findings demonstrated that any CLIA result of $\leq 3.0$ RLU should be considered a negative result while a CLIA result of $>10.0$ RLU should define a positive result. In additional, were analyzed diagnostic Performance of Immunological Assays for the Detection of HTLV-1 and HTLV-2 Infection (**S2 Table**).

In order to accomplish the differential diagnosis of HTLV-1 and HTLV-2 infection, the WB and the FC-Duplex assay were carried out in parallel batches. The WB results for HTLV-1 from HTLV-2 for differential diagnosis were considered according to the manufacturer instructions and the FC-Duplex results were classified using the criteria previously proposed by Pimenta de Paiva et al. [28]. According to their criteria, the differential diagnosis of HTLV infection using FC-Duplex can be achieved as the $\Delta$PPFC = (MT-2 1:32—Mot 1:32). Using this criterion, out of 22 of the tested samples, fifteen were classified as HTLV-1 and seven as HTLV-2. Comparative analysis between the WB and FC-Duplex results (criterion 1) demonstrated 100% agreement for HTLV-1 (15/15) and HTLV-2 (7/7) (**Fig 3**). Pimenta de Paiva et al. [28] proposed two additional criteria for differential diagnosis of HTLV infection using FC-Duplex. The use of these criteria did not show outstanding agreement when compared with the WB reference diagnosis [criterion 2 (HTLV-1 (15/15), HTLV-2 (4/7) with 3/7 misclassifications); criterion 3 (HTLV-1 (15/15), HTLV-2 (5/7) with 2/7 misclassifications)] (**S1 and S2 Figs**).

## 4. Discussion

During the 5 years of screening blood donors that were reactive for HTLV-1/2 at the HEMOAM Foundation, we detected 409 individuals who were considered unfit to donate due to reactive serology for the virus when using the CLIA. At the time of the donation attempt, these individuals are flagged in the HEMOSYS system as having pending issues. They are subsequently contacted by the blood center and asked to return for a retest to confirm the diagnosis. This procedure occurs in accordance with guidance of the Brazilian Ministry of Health and Law 17,344, which makes the diagnosis and monitoring of individuals infected with HTLV mandatory, so that they can be treated for their serological condition [25].

During the study period (2018 to 2022), we identified a prevalence of 0.15% of anti-HTLV antibodies (n = 257,942 eligible donors during the study period). We observed a slight increase in this prevalence when we compared it with data from previous surveys carried out in the same region of Brazil, which identified a prevalence of 0.13% (n = 87,402) in the period from 2001 to 2003, and 0.14% (n = 6,865) in the period from 2008 to 2009 when using ELISA. This indicates that the prevalence in the state was generally maintained [23, 37]. It is important to consider that the present study presented a longer survey period and covered a greater number of donors, though a similar prevalence was observed. These data demonstrate that Amazonas continues to be one of the states with the lowest prevalence when compared to others in the northern region of Brazil, such as Amapá (0.71%) and Pará (0.91%) [24, 38].

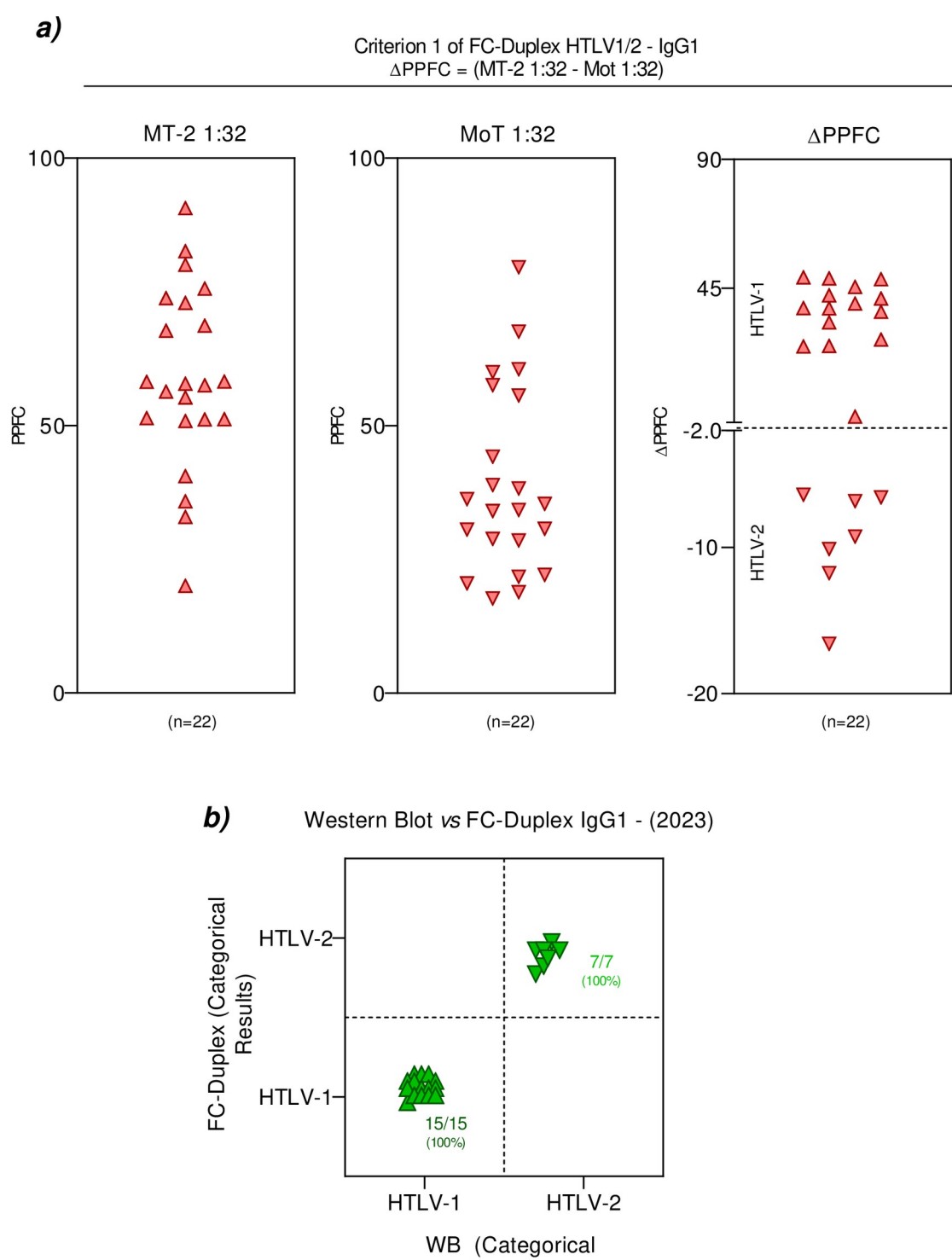

**Fig 3. Differential diagnosis of HTLV-1/2 infection after retesting using the FC-Duplex IgG1.** *a)* Anti- MT-2 (1:32) and anti-MoT (1:32) reactivity used to define the HTLV-1 and HTLV-2 diagnosis according to criterion 1, previously proposed by Pimenta de Paiva et al. [28]. *b)* Agreement between FC-Duplex HTLV-1/2 IgG1 and western blot results. *PPFC: percentage of positive fluorescent cells; WB: western blot assay.*

Despite the low prevalence of HTLV-reactive blood donors in the city of Manaus, new studies must be carried out, especially considering other populations that are not blood donors, such as riverine and indigenous people. Conducting research with other populations would provide complementary data and give a better overview of the prevalence of the HTLV virus in Amazonas. If new research is carried out on blood donors, it would be interesting to use the same assay methodologies as in the present study, with the aim of providing a safer comparison, taking into account that ELISA and CLIA may differ in terms of the screening and diagnosis process [15, 33, 38, 39].

In our study, of the 151 individuals resubmitted to the CLIA, 93 (62%) were positive and were selected to undergo confirmatory retests. Using the WB, the main methodology used to confirm the type of HTLV, we identified 30 (20%) positive, 10 (7%) indeterminate and 111 (73%) negative samples. In addition, via the FC-Simplex, 26 (29%) positive and 67 (71%) negative samples for HTLV were identified. Indeterminate cases in the WB assay may be associated with the individual's seroconversion period, as well as cross reactions, which make accurate diagnosis of HTLV difficult [40]. In addition, inconclusive results in confirmatory serological assays may be associated with other factors, such as the manufacturer's strict positivity criteria, HTLV strains different from those used in the tests, presence of viral particles and point mutations in the long terminal repeat viral promoter region and regions that encode structural proteins [14, 15, 30]. Cross-reactivity may be linked to individuals who had a recent *P. falciparum* infection, as studies in endemic areas have shown that infection by the parasite can interfere with the result and is capable of producing false positives in the WB [41]. Noteworthy, the Amazon region of Brazil is considered an endemic area for malaria in the country, with 99% of autochthonous cases, and this could contribute to the number of indeterminate cases observed [38, 42].

Regarding the comparison of the assays evaluated, when we categorized them into low, intermediate and high RLUs (**Fig 2A**) according to the CLIA sensitivity cut-off point, it was identified that the low and intermediate zones were negative in the WB and the FC-Simplex. The CLIA's very low cut-off point is a good factor for using this methodology as a screening test, as has been observed in other studies carried out to verify the application of this assay during screening [33–35, 39]. However, false-positive results in immunological assays need to be evaluated and minimized, as this impacts the quality of life of candidates for blood donations with a positive result during screening tests. In our study, we propose that the RLU values currently used in the CLIA need to be revised, and any CLIA results of ≤3.0 RLU should be considered negative results while CLIA results of >10.0 RLU would define a positive result. Other studies reinforce these findings and state that the assays need to be revised, and sequential tests can be used with different methodologies or adjustments in the cutoff values in the CLIA used to screen blood donors in Brazil, thus avoiding the use of confirmatory assays [33, 40, 43, 44].

Despite the CLIA's weaknesses in relation to the seroconversion period of individuals, in addition to cross-reactivity, the test appears to be efficient in preventing the passage and distribution of blood bags contaminated with the HTLV virus. However, it does not have the same efficiency when used as a diagnostic method for the virus, due to its low agreement (56%) when compared to the WB. On the other hand, the FC-Simplex assay (still in the validation phase) showed better performance (agreement = 80%), and can be used to confirm the screening process, before confirming the differential diagnosis with the WB or qPCR [29]. Additionally, we observed that the FC-Duplex, the assay responsible for the viral distinction between HTLV-1 and HTLV-2 as a phase after FC-Simplex, also presented excellent results, with 100% agreement when compared to the WB for the samples analyzed [28, 29]. Although these results are promising, we note that only criterion 1 (MT-2 1;32 minus MoT 1;32) was considered satisfactory for agreement with the WB results, whereby 15 (100%) reactive samples were

identified for HTLV-1 and 7 (100%) for HTLV-2. This highlights the ability of the FC-Duplex to distinguish other viruses from HTLV, while also being a low cost and accessible assay for blood banks in Brazil [28].

New studies have been carried out to evaluate the sensitivity and specificity of immunological and molecular assays for screening HTLV infection in the screening of people living with HIV, during prenatal care of pregnant women, as well as sexually transmitted co-infections, such as hepatitis A virus (HAV), HBV, HCV, syphilis, chlamydia and gonorrhea [14, 45]. Furthermore, immunological assays using the immunochromatographic method for detection of anti-HTLV antibodies have been developed and validated for rapid screening of HTLV-1/2 infection in different diagnostic situations, in addition to evaluating cross-reaction to other pathogens [46, 47].

It is important to highlight that our study has some limitations. In the initial survey of individuals that were reactive for HTLV, 409 blood donors that were reactive for the virus were identified. However, due to the lack of updating of records in the institution's database, it was not possible to obtain contact with a significant number of individuals, and a large portion did not show interest in returning for a new CLIA, which ended up having a significant impact on the number of participants in the study. In addition, we did not perform the qPCR technique, which would have provided valuable data for comparison with other methodologies, in addition to being useful to elucidate the serological status of the ten samples that were categorized as indeterminate in the WB. Finally, due to the state of Amazonas being an endemic area for malaria, it is important to evaluate possible co-infection in these individuals so as to eliminate possible biases.

## 5. Conclusion

Our results demonstrate the low seroprevalence of HTLV among candidates for blood donation in the state of Amazonas. They also demonstrate that levels are lower than those of other states in the northern region of Brazil. We also showed that the CLIA is highly effective in the screening stage of candidates for blood donations, although it presents many false-positive results and the need for confirmatory tests. In addition, FC-simplex and FC-Duplex were able to identify reactive samples and could make the distinction between HTLV-1/2. It is therefore a promising methodology for confirming HTLV infection when compared with WB, which is the reference assay. Furthermore, our findings indicate that gaps in the diagnosis of HTLV-1/2 infection could be overcome by the simultaneous use of distinct immunological assays during retesting of candidates for blood donations.

## Supporting information

**S1 File. Immunological assays methods.**
(DOCX)

**S1 Table. Demographical and laboratorial records of candidates for blood donations with positive results for HTLV-1/2 after serological retesting using CLIA.**
(DOCX)

**S2 Table. Diagnostic performance of immunological assays for the detection of HTLV-1 and HTLV-2 infection.**
(DOCX)

**S1 Fig. Differential diagnosis of HTLV-1/2 infection after retesting using the FC-Duplex IgG1.** *a)* Anti-MT-2 (1:32) and anti-MoT (1:1,024) reactivity used to define the HTLV-1 and HTLV-2 diagnosis according to criterion 1, previously proposed by Pimenta de Paiva et al.

[28]. *b)* Agreement between FC-Duplex HTLV-1/2 IgG1 and western blot results. *PPFC*: *percentage of positive fluorescent cells; WB*: *western blot assay.*
(PDF)

**S2 Fig. Differential diagnosis of HTLV-1/2 infection after retesting using the FC-Duplex IgG1.** *a)* Anti- MT-2 (1:32) and anti-MoT (1:2,048) reactivity used to define the HTLV-1 and HTLV-2 diagnosis according to criterion 1, previously proposed by Pimenta de Paiva et al. [28]. *b)* Agreement between FC-Duplex HTLV-1/2 IgG1 and western blot results. *PPFC*: *percentage of positive fluorescent cells; WB*: *western blot assay.*
(PDF)

## Author Contributions

**Conceptualization:** Felipe Araujo Santos, Andréa Teixeira-Carvalho, Márcio Sobreira Silva Araújo, Olindo Assis Martins-Filho, Allyson Guimarães Costa.

**Data curation:** Nilberto Dias Araújo, Fábio Magalhães-Gama, Andréa Teixeira-Carvalho, Gemilson Soares Pontes, Márcio Sobreira Silva Araújo, Olindo Assis Martins-Filho.

**Formal analysis:** Felipe Araujo Santos, Fábio Magalhães-Gama, Vanessa Peruhype-Magalhães, Jordana Grazziela Alves Coelho-dos-Reis, Andréa Teixeira-Carvalho, Antonio Carlos Rosário Vallinoto, Gemilson Soares Pontes, Márcio Sobreira Silva Araújo, Olindo Assis Martins-Filho, Allyson Guimarães Costa.

**Funding acquisition:** Antonio Carlos Rosário Vallinoto, Gemilson Soares Pontes, Olindo Assis Martins-Filho, Allyson Guimarães Costa.

**Investigation:** Cláudio Lucas Santos Catão, Júlia Pereira Martins, Uzamôr Henrique Soares Pessoa, Isabelle Vasconcelos Sousa, Jean Silva Melo, Gláucia Lima Souza, Nilberto Dias Araújo, Fábio Magalhães-Gama, Cláudia Maria de Moura Abrahim, Emmily Myrella Vasconcelos Mourão.

**Methodology:** Felipe Araujo Santos, Cláudio Lucas Santos Catão, Júlia Pereira Martins, Uzamôr Henrique Soares Pessoa, Isabelle Vasconcelos Sousa, Jean Silva Melo, Gláucia Lima Souza, Nilberto Dias Araújo, Fábio Magalhães-Gama, Cláudia Maria de Moura Abrahim, Emmily Myrella Vasconcelos Mourão, Márcio Sobreira Silva Araújo.

**Project administration:** Olindo Assis Martins-Filho, Allyson Guimarães Costa.

**Resources:** Júlia Pereira Martins, Isabelle Vasconcelos Sousa, Jean Silva Melo, Gláucia Lima Souza.

**Supervision:** Jordana Grazziela Alves Coelho-dos-Reis.

**Writing – original draft:** Felipe Araujo Santos, Nilberto Dias Araújo, Márcio Sobreira Silva Araújo, Olindo Assis Martins-Filho.

**Writing – review & editing:** Vanessa Peruhype-Magalhães, Jordana Grazziela Alves Coelho-dos-Reis, Andréa Teixeira-Carvalho, Antonio Carlos Rosário Vallinoto, Gemilson Soares Pontes, Allyson Guimarães Costa.

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
