## [Decision Letter · Decision Letter 0]

4 Mar 2024

PONE-D-24-03307Performance of Immunological Assays for Universal and Differential Diagnosis of HTLV-1/2 Infection in Candidates for Blood Donations from the Brazilian AmazonPLOS ONE

Dear Dr. Costa,

Thank you for submitting your manuscript to PLOS ONE. After careful consideration, we feel that it has merit but does not fully meet PLOS ONE’s publication criteria as it currently stands. Therefore, we invite you to submit a revised version of the manuscript that addresses the points raised during the review process.

Please address ALL review comments and thoroughly revise the manuscript accordingly in order to make the submission suitable for publication:

Critical comments include:

a) Please provide the complete protocols of the methods used as supplementary materials.

b) Since the manuscript is on diagnostic immulogical assays of HTLV, the flow cytometry-based immunoassay is commendable, however, why did the authors not consider INNOLIA in the study? INNOLIA and WB are confirmatory and would specifically prove the presence of HTLV.

c) Did authors use a different CLIA kit a second time? Why CLIA retest? Additionally, the high disparity of outcome upon assay repeat is questionable. 

d) How did they authors analyse the WB blots? This was not demonstrated.

e) Please provide all raw results (blots etc) as supplementary data.

We look forward to receiving your revised manuscript.

Kind regards,

Nneoma Confidence JeanStephanie Anyanwu, Ph.D.

Academic Editor

PLOS ONE

Additional Editor Comments:

Please address ALL review comments and thoroughly revise the manuscript accordingly in order to make the submission suitable for publication:

Critical comments include:

a) Please provide the complete protocols of the methods used as supplementary materials.

b) Since the manuscript is on diagnostic immulogical assays of HTLV, the flow cytometry-based immunoassay is commendable, however, why did the authors not consider INNOLIA in the study? INNOLIA and WB are confirmatory and would specifically prove the presence of HTLV.

c) Did authors use a different CLIA kit a second time? Why CLIA retest? Additionally, the high disparity of outcome upon assay repeat is questionable.

d) How did they authors analyse the WB blots? This was not demonstrated.

e) Please provide all raw results (blots etc) as supplementary data.

Reviewers' comments:

Reviewer's Responses to Questions

**Comments to the Author**

1. Is the manuscript technically sound, and do the data support the conclusions?

Reviewer #1: Yes

Reviewer #2: Yes

Reviewer #3: Partly

Reviewer #4: Yes

Reviewer #5: Yes

2. Has the statistical analysis been performed appropriately and rigorously? 

Reviewer #1: Yes

Reviewer #2: Yes

Reviewer #3: No

Reviewer #4: No

Reviewer #5: Yes

3. Have the authors made all data underlying the findings in their manuscript fully available?

Reviewer #1: Yes

Reviewer #2: Yes

Reviewer #3: Yes

Reviewer #4: Yes

Reviewer #5: Yes

4. Is the manuscript presented in an intelligible fashion and written in standard English?

Reviewer #1: Yes

Reviewer #2: Yes

Reviewer #3: No

Reviewer #4: Yes

Reviewer #5: Yes

5. Review Comments to the Author

Reviewer #1: The article is very interesting and important for follow.up the oncogenic viruses. I hope there was sequencing for the strains and detection of the possibility of cancer production so they could follow up these cases for any tumour appearance

Reviewer #2: This manuscript compares the ability of 3 immunological assays for the detection of anti-HTLV antibodies in candidates for blood donations in Brazil.

1. Manuscript is well-written and the flow is easy to follow.

2. The figure caption for "supplementary table 1" is missing or move the caption to the corresponding supplementary figure 1 and 2.

3. Line 135 sentence isn't complete.

4. Overall this manuscript addresses the need for retesting for a complete and accurate diagnosis of HTLV. Authors included limitations and the possibility of co-infection in the area endemic for malaria.

Reviewer #3: 1- Why the authors did not consider any NAT techniques? As they used the flowcytometry methods the cost of assessment will not differ. While, the NAT method can prove the infection.

2- What is the difference between CLIA screening test and CLIA retest? If both of them are the same why the CLIA retest did preformed and how the positive samples lost its positivity for around 40%? While the total allowable error and the respected CV% for CLIA should be less than 7% and 3% respectively.

3- Your only second CLIA positive results (93 samples) consider for more evaluation while the complementary tests did not perform on total 151 samples.

4- What was the first cut off value for positive consideration of samples in CLIA screening test?

5- The authors cannot provide the distinct results because each method has different diagnostic specify and sensitivity and none of samples were proved by NAT assay.

Reviewer #4: The present study compares the ability of distinct immunological assays (chemiluminescence immunoassay-CLIA, western blot-WB and flow cytometry-FC Simplex and Duplex) to detect anti-HTLV antibodies in candidates for blood donations at the Amazonas State Blood Center (Brazil) between January 2018 and December 2022. The authors suggest that gaps in the diagnosis of infection with HTLV-1/2 could be overcome by the simultaneous use of distinct immunological assays during retesting of candidates for blood donations.

Comments:

1. Please explain the abbreviation, HTLV when used first time in the abstract and the main text. Abbreviations for various techniques used could also be explained in the main body of the article when used for the first time.

2. The widespread and epidemiology of the disease is convincedly covered in the introduction section, however the impact and severity of the disease is not plausibly and convincingly covered in the introduction section. A broad mechanism and deleterious effect of the disease could be explained in the light of deleterious effect of the disease.

3. Review and update the references to encompass the latest available literature. It's important to enhance the description by incorporating recent studies related to the topic.

4. Consider including a table summarizing/comparing the utilization and results of CLIA vs WB, CLIA vs FC.

5. The statistical methods employed must be explained properly.

6. All the figure legends must describe the figure appropriately and plausibly including the statistical methods.

Reviewer #5: The authors in this manuscript have compared CLIA, western blot (WB), and flow cytometry (FC) Simplex and Duplex assays to detect anti-HTLV antibodies in the blood plasma samples of blood donation candidates for the Amazonas State Blood Center in Brazil. They have also suggested that CLIA results of ≤3 RLU should be considered a negative result. This work focuses on the clinically important aspect of considering reliable diagnostic methods for screening HTLV-negative blood donors. The manuscript is well-written. While reviewing the manuscript, I found a few mistakes in the writing, and I have a few questions and concerns which are listed below:

1. The authors have suggested considering <3 RLU as a negative result. While analyzing, they also found that 58 samples, which were positive at an earlier time point but <1 RLU, became negative by the same assay at a later time point. Did the authors check these samples by any other assays to confirm whether these samples were negative for HTLV by other assays as well? Also, I believe there might be other companies selling HTLV CLIA kits, and there could be batch-to-batch variations too. Do they know that all of these kits have the same cutoff and sensitivity? The purpose of screening is to prevent infection by blood transfusion. Unless the cutoff is universal, don’t they think such a suggestion could be risky?

2. The authors have suggested that Amazon is endemic for malaria and the intermediate category in western blot could be because of antibodies against Plasmodium proteins. RLU for those samples is over 3 so according to their CLIA cutoff, it could be positive. Because of EDTA blood, RT-PCR can be an issue, but can they actually prove by ELISA using Plasmodium lysate?

3. Also, in their discussion section (lines 271-273), the explanation of seroconversion doesn’t seem justified because they enrolled the subjects who were found to be seropositive much earlier, and it was the basis of their enrollment in the study.

4. Are the CLIA positive FC-Simplex negative samples the same as the WB intermediate and positive FC-Simplex negative in figure 2b?

5. Line 135 seems incomplete.

6. Line 146 has an error for Alinity's. There could be other grammatical mistakes too but I didn't focus much on that.

6. PLOS authors have the option to publish the peer review history of their article (what does this mean?). If published, this will include your full peer review and any attached files.

Reviewer #1: **Yes: **Michael Nazmy Agban professor of Microbiology and Immunology faculty of medicine assiut university Egypt

Reviewer #2: No

Reviewer #3: No

Reviewer #4: **Yes: **Mohammad Asad

Reviewer #5: No

---

## [Author Response · Author response to Decision Letter 0]

8 May 2024

Nneoma Confidence JeanStephanie Anyanwu, Ph.D.

Academic Editor

PLOS ONE

Manaus, May 08th, 2024

Dear Editor,

We very much appreciate the kind consideration given to our manuscript (MS) entitled "Performance of Immunological Assays for Universal and Differential Diagnosis of HTLV-1/2 Infection in Candidates for Blood Donations from the Brazilian Amazon". We hope that the replies to the reviewer’s comments will have satisfactorily improved the MS.

Below, we present all the queries made by the reviewers. The changes requested are clearly outlined in the revised manuscript and marked in yellow. We have prepared a list of answers to the reviewers’ comments, which are highlighted in “bold italic”. In the response to each query, we are also including the modified part, as it is in the revised manuscript.

Editor comments:

1. Please provide the complete protocols of the methods used as supplementary materials.

We acknowledge the comment of Editor and inform that complete protocols were inserted in supplementary materials.

2. Since the manuscript is on diagnostic immulogical assays of HTLV, the flow cytometry-based immunoassay is commendable, however, why did the authors not consider INNOLIA in the study? INNOLIA and WB are confirmatory and would specifically prove the presence of HTLV.

We acknowledge the comment of Editor and inform that for this study we used the HTLV blot 2.4 kit (MP diagnostics®) as a Western Blot immunoassay, as there were problems importing the INNO-LIA® HTLV I/II Ab Score. In additional, HTLV blot 2.4 kit is qualitative enzyme immunoassay uses for detection of the antibodies to HTLV-1/2 in human serum and is used as a confirmatory step methodology that is capable of detecting and distinguishing virus types 1 and 2, as recommended by the Food and Drug Administration (FDA) and Brazilian Ministry of Health.

3. Did authors use a different CLIA kit a second time? Why CLIA retest? Additionally, the high disparity of outcome upon assay repeat is questionable.

We acknowledge the comment of Editor and inform that a different CLIA test was not used in the retest. In both screening tests, the chemiluminescence assay (CLIA) with Alinity s rHTLV-I/II kit (Abbott®) was used. As for the retest, this was carried out following the recommendation of the guidance of the Brazilian Ministry of Health and Law 17,344, which makes the diagnosis and monitoring of individuals infected with HTLV mandatory, so that they can be treated for their serological condition [doi:10.1590/s1679 -497420200006000015.esp1]. Finally, the high disparity between the results obtained in the 1st and 2nd test may be related to the cross-reactivity the CLIA assay presents and described in other studies [doi:10.1111/j.1365-3148.2009.00932.x, doi:10.1111/tme.12482; doi:10.1002/jcla.22909; doi:10.1111/j.1537-2995.2009.02572.x], besides cross-reaction with HIV-I, HIV-II, hepatitis A virus, hepatitis B virus, hepatitis C virus, herpes simplex virus, Epstein–Barr virus, SARS-CoV-2, Chlamydia trachomatis, Neisseria gonorrhoeae, Treponema pallidum, Toxoplasma gondii, and Plasmodium falciparum [doi: 10.3390/v15010129].

4. How did they authors analyse the WB blots? This was not demonstrated.

We acknowledge the comment of Editor and inform that the text with analyze the WB blots and the requested information has been included (Line 160-169). 

5. Please provide all raw results (blots etc) as supplementary data.

We acknowledge the comment of Editor and inform that included WB results in supplementary materials.

 

Reviewers' comments:

Reviewer reports:

Reviewer #1: 

1. The article is very interesting and important for follow.up the oncogenic viruses. I hope there was sequencing for the strains and detection of the possibility of cancer production so they could follow up these cases for any tumour appearance

We acknowledge the comment of Reviewer #1 and inform that we are optimistic to know that our work is well-designed and will serve as a basis for new prospective studies. Furthermore, we inform that we have carried out work in parallel with the molecular analysis of HTLV subtypes.

Reviewer #2: 

1. This manuscript compares the ability of 3 immunological assays for the detection of anti-HTLV antibodies in candidates for blood donations in Brazil.

2. Manuscript is well-written and the flow is easy to follow.

We acknowledge the comment of Reviewer #2 and inform you that we are optimistic to know that our work is well-designed.

3. The figure caption for "supplementary table 1" is missing or move the caption to the corresponding supplementary figure 1 and 2.

We acknowledgment the comment of Reviewer #2 and we inform you that we include figure caption in supplementary figure 1 and 2 files. 

4. Line 135 sentence isn't complete.

We acknowledgment the comment of Reviewer #2 and we inform you that we rewrite the sentence (line 138-141). 

5. Overall this manuscript addresses the need for retesting for a complete and accurate diagnosis of HTLV. Authors included limitations and the possibility of co-infection in the area endemic for malaria.

We acknowledge the comment of Reviewer #2 and inform you that we are optimistic to know that our work is well-designed.

Reviewer #3:

1. Why the authors did not consider any NAT techniques? As they used the flowcytometry methods the cost of assessment will not differ. While, the NAT method can prove the infection.

We acknowledge the comment of Reviewer #3 and inform that we did not use a NAT technique because this is a study evaluating different immunological methods for detecting antibodies and that this could generate a bias in the analysis of the results of immunological assays.

2. What is the difference between CLIA screening test and CLIA retest? If both of them are the same why the CLIA retest did preformed and how the positive samples lost its positivity for around 40%? While the total allowable error and the respected CV% for CLIA should be less than 7% and 3% respectively.

We acknowledge the comment of Reviewer #3 and inform that there is no difference between the tests performed, except for the time between 1st and 2nd test. In both screening tests, the chemiluminescence assay (CLIA) with Alinity s rHTLV-I/II kit (Abbott®) was used. In additional, the high disparity between the results obtained in the 1st and 2nd test may be related to the cross-reactivity the CLIA assay presents and described in other studies [doi:10.1111/j.1365-3148.2009.00932.x, doi:10.1111/tme.12482; doi:10.1002/jcla.22909; doi:10.1111/j.1537-2995.2009.02572.x], besides the cross-reaction with HIV-I, HIV-II, HAV, HBV, HCV, herpes simplex virus, Epstein–Barr virus, SARS-CoV-2, Chlamydia trachomatis, Neisseria gonorrhoeae, Treponema pallidum, Toxoplasma gondii, and Plasmodium falciparum [doi: 10.3390/v15010129].

3. Your only second CLIA positive results (93 samples) consider for more evaluation while the complementary tests did not perform on total 151 samples.

We acknowledge the comment of Reviewer #3 and inform that the negative samples in the 2nd CLIA assays test were monitored by the blood bank team. In additional, this was carried out following the recommendation of the guidance of the Brazilian Ministry of Health and Law 17,344, which makes the diagnosis and monitoring of individuals infected with HTLV mandatory, so that they can be treated for their serological condition [doi:10.1590/s1679 -497420200006000015.esp1].

4. What was the first cut off value for positive consideration of samples in CLIA screening test?

We acknowledge the comment of Reviewer #3 and inform that the text with cut off value for positive results in CLIA assays has been included (Line 156-157). 

5. The authors cannot provide the distinct results because each method has different diagnostic specify and sensitivity and none of samples were proved by NAT assay.

We acknowledge the comment of Reviewer #3 and inform that we include the results obtained in each immunological assay, being aware that each method has different diagnostic specification and sensitivity, although they all have the same target: detection of anti-HTLV-1/2 antibodies.

Reviewer #4: 

The present study compares the ability of distinct immunological assays (chemiluminescence immunoassay-CLIA, western blot-WB and flow cytometry-FC Simplex and Duplex) to detect anti-HTLV antibodies in candidates for blood donations at the Amazonas State Blood Center (Brazil) between January 2018 and December 2022. The authors suggest that gaps in the diagnosis of infection with HTLV-1/2 could be overcome by the simultaneous use of distinct immunological assays during retesting of candidates for blood donations.

We acknowledge the comment of Reviewer #4 and inform that we are optimistic to know that our work is well-designed and will serve as a basis for new prospective studies

1. Please explain the abbreviation, HTLV when used first time in the abstract and the main text. Abbreviations for various techniques used could also be explained in the main body of the article when used for the first time.

We acknowledgment the comment of Reviewer #4 and we inform that we revised explain the abbreviations in manuscript. 

2. The widespread and epidemiology of the disease is convincedly covered in the introduction section, however the impact and severity of the disease is not plausibly and convincingly covered in the introduction section. A broad mechanism and deleterious effect of the disease could be explained in the light of deleterious effect of the disease.

We acknowledgment the comment of Reviewer #4 and we inform that we included the information in manuscript (line 70-79). 

3. Review and update the references to encompass the latest available literature. It's important to enhance the description by incorporating recent studies related to the topic.

We acknowledgment the comment of Reviewer #4 and we inform that we performed review and update the references in manuscript. 

4. Consider including a table summarizing/comparing the utilization and results of CLIA vs WB, CLIA vs FC.

We acknowledgment the comment of Reviewer #4 and we inform that we included the information in Supplementary Table 1 and 2.

5. The statistical methods employed must be explained properly.

We acknowledgment the comment of Reviewer #4 and we inform that we revised the “Statistical analysis” topic and included new information’s about statistical methods included. 

6. All the figure legends must describe the figure appropriately and plausibly including the statistical methods.

We acknowledgment the comment of Reviewer #4 and we inform that we performed review and update the figure legends. 

Reviewer #5: 

The authors in this manuscript have compared CLIA, western blot (WB), and flow cytometry (FC) Simplex and Duplex assays to detect anti-HTLV antibodies in the blood plasma samples of blood donation candidates for the Amazonas State Blood Center in Brazil. They have also suggested that CLIA results of ≤3 RLU should be considered a negative result. This work focuses on the clinically important aspect of considering reliable diagnostic methods for screening HTLV-negative blood donors. The manuscript is well-written. While reviewing the manuscript, I found a few mistakes in the writing, and I have a few questions and concerns which are listed below: The authors have suggested considering <3 RLU as a negative result. While analyzing, they also found that 58 samples, which were positive at an earlier time point but <1 RLU, became negative by the same assay at a later time point. Did the authors check these samples by any other assays to confirm whether these samples were negative for HTLV by other assays as well? Also, I believe there might be other companies selling HTLV CLIA kits, and there could be batch-to-batch variations too. Do they know that all of these kits have the same cutoff and sensitivity? The purpose of screening is to prevent infection by blood transfusion. Unless the cutoff is universal, don’t they think such a suggestion could be risky?

We acknowledgment the comment of Reviewer #5 and we inform that we used Alinity's rHTLV-I/II kit (Abbott®) as it is provided by the Brazilian Ministry of Health. As for the kits used, they have the same cutoff value, according to information entered in the methods (line: 156-157). As for our proposal to increase the cutoff value, this was suggested, as one of the items to be evaluated when there are many false-positive results is the stringent manufacturer positivity criteria (doi: 10.1590/0037-8682-0175-2021). This review of the kit supplier can help blood banks reduce the disposal of bags that show false-positive results, minimizing costs for blood banks.

1. The authors have suggested that Amazon is endemic for malaria and the intermediate category in western blot could be because of antibodies against Plasmodium proteins. RLU for those samples is over 3 so according to their CLIA cutoff, it could be positive. Because of EDTA blood, RT-PCR can be an issue, but can they actually prove by ELISA using Plasmodium lysate?

We acknowledgment the comment of Reviewer #5 and we inform that in new studies, the presence of anti-Plasmodium antibodies in these samples will be evaluated in order to rule out this hypothesis.

2. Also, in their discussion section (lines 271-273), the explanation of seroconversion doesn’t seem justified because they enrolled the subjects who were found to be seropositive much earlier, and it was the basis of their enrollment in the study.

We acknowledgment the comment of Reviewer #5 and inform that we included new possibilities for the presence of false-positive results in the analyzed samples (line 294-300).

3. Are the CLIA positive FC-Simplex negative samples the same as the WB intermediate and positive FC-Simplex negative in figure 2b?

We acknowledgment the comment of Reviewer #5 and inform that yes, they are the same samples. We will now carry out further studies with these samples to clarify the factors involved in the cross-reaction that was observed in our manuscript.

4. Line 135 seems incomplete.

We acknowledgment the comment of Reviewer #5 and we inform that we revised the in manuscript.

5. Line 146 has an error for Alinity's. There could be other grammatical mistakes too but I didn't focus much on that.

We acknowledgment the comment of Reviewer #5 and we inform that we revised the in manuscript.

We believed that the new changes have significantly improved the quality of our manuscript. We would like to thank Plos One members and reviewers for their dedication in providing their valuable and interesting comments on this article. 

We sincerely hope that the revised version of our manuscript meets the high standards of Plos One publication’s and is, therefore, acceptable for publication in this journal.

Best regards,

Allyson Guimarães da Costa (on behalf of all authors)

---

## [Decision Letter · Decision Letter 1]

12 Jun 2024

Performance of Immunological Assays for Universal and Differential Diagnosis of HTLV-1/2 Infection in Candidates for Blood Donations from the Brazilian Amazon

PONE-D-24-03307R1

Dear Dr. Costa,

We’re pleased to inform you that your manuscript has been judged scientifically suitable for publication and will be formally accepted for publication once it meets all outstanding technical requirements.

Kind regards,

Nneoma Confidence JeanStephanie Anyanwu, Ph.D.

Academic Editor

PLOS ONE

Additional Editor Comments (optional):

Reviewers' comments:

Reviewer's Responses to Questions

**Comments to the Author**

1. If the authors have adequately addressed your comments raised in a previous round of review and you feel that this manuscript is now acceptable for publication, you may indicate that here to bypass the “Comments to the Author” section, enter your conflict of interest statement in the “Confidential to Editor” section, and submit your "Accept" recommendation.

Reviewer #4: All comments have been addressed

Reviewer #5: All comments have been addressed

2. Is the manuscript technically sound, and do the data support the conclusions?

Reviewer #4: Yes

Reviewer #5: Partly

3. Has the statistical analysis been performed appropriately and rigorously? 

Reviewer #4: I Don't Know

Reviewer #5: Yes

4. Have the authors made all data underlying the findings in their manuscript fully available?

Reviewer #4: Yes

Reviewer #5: Yes

5. Is the manuscript presented in an intelligible fashion and written in standard English?

Reviewer #4: Yes

Reviewer #5: Yes

6. Review Comments to the Author

Reviewer #4: The authors have addressed the comments and it could be published if the editor and other reviewers are convinced

Reviewer #5: The authors have addressed most of the questions and concerns raised by me previously. The manuscript looks good in its current form though I haven't focused much on the grammatical errors.

7. PLOS authors have the option to publish the peer review history of their article (what does this mean?). If published, this will include your full peer review and any attached files.

Reviewer #4: **Yes: **Mohammad Asad

Reviewer #5: **Yes: **Akil Akhtar

---

## [Editor Report · Acceptance letter]

26 Jun 2024

PONE-D-24-03307R1 

PLOS ONE

Dear Dr. Costa, 

I'm pleased to inform you that your manuscript has been deemed suitable for publication in PLOS ONE. Congratulations! Your manuscript is now being handed over to our production team.

Kind regards, 

on behalf of

Dr. Nneoma Confidence JeanStephanie Anyanwu 

Academic Editor

PLOS ONE